

# Assessment of climate change and land use development effects on dam reliability

P. Taherei Ghazvinei[1,2], H. Hassanpour Darvishi[1], R. B. Hashim[3], S. H. Musavi Jahromi[4], and N. Aghamohammadi[5]

[1]Department of Water Science and Engineering, Shar-e-Qods Branch, Islamic Azad university (IAU), Tehran, Iran
[2]Young Researchers and Elite Club, Shahr-e-Qods Branch, Islamic Azad University, Tehran, Iran
[3]Department of Civil Engineering, Faculty of Engineering, University Of Malaya, Kuala Lumpur, Malaysia
[4]Civil Engineering Department, Technical and Engineering College, Shahr-e-Qods Branch, Islamic Azad University (IAU), Tehran, Iran
[5]Center for Occupational and Environmental Health, Department of Social and Preventive Medicine, University of Malaya, Kuala Lumpur, Malaysia

Discussion Paper | Discussion Paper | Discussion Paper | Discussion Paper

HESSD

doi:10.5194/hess-2015-481

Cclimate change and land use development effects on dam reliability

P. Taherei Ghazvinei et al.

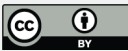

Received: 5 November 2015 – Accepted: 15 November 2015 – Published: 21 January 2016

Correspondence to: P. Taherei Ghazvinei (p.taherei@gmail.com)

Published by Copernicus Publications on behalf of the European Geosciences Union.

Discussion Paper | Discussion Paper | Discussion Paper | Discussion Paper |

**HESSD**

doi:10.5194/hess-2015-481

**Cclimate change and land use development effects on dam reliability**

P. Taherei Ghazvinei et al.

## Abstract

The purpose of this study was to assess long-term impacts of climate and land use change on a catchment runoff and dam overtopping dam reliability. Long hydrological time series (30 years) from six rainfall stations and one stream flow stations were anal-
⁵ ysed. A methodology combining common statistical methods with hydrological modelling was adopted in order to distinguish between the effects of climate and land use change and to present probabilistic assessment of overtopping reliability of the selected earth-fill dam. It is important to ensure that extreme meteorologically induced flood rises do not exceed dam crest level. Considering climate change factor, intensity-
¹⁰ duration-frequency curves of the catchment were updated. In addition, in consistency with the areal development plan, year of 2020 was targeted to evaluate the effect of land use changes on the generation of storm runoff. Accordingly, compared with current imperviousness it was found that the areal imperviousness will be increased up to 4.5 % by the year of 2030. Step-by-step procedures were carried out in tandem to eval-
¹⁵ uate the hydrological performance of the spillway capacity in light of an extreme storm event of PMP/PMF magnitude. The HEC-HMS was applied to transform the PMPs to PMFs and estimate the outflows and corresponding flood rises over the crest level for all durations. A conventional reservoir routing procedure with modified technique was then carried out for all PMP/PMF durations; i.e., 1 to 120 h. Three cases of differ-
²⁰ ent floods were performed where the last case represented the most severe flood on the dam reservoir. The simulations indicated that the flood rises for all durations were lower than the embankment crest level. Although results showed marginally adequate capacity to allow safe passage of flood water of PMP/PMF magnitude, the dam safety in hydrological aspect was assured.

Discussion Paper | Discussion Paper | Discussion Paper | Discussion Paper | Discussion Paper

**HESSD**

doi:10.5194/hess-2015-481

**Cclimate change and land use development effects on dam reliability**

P. Taherei Ghazvinei et al.

Discussion Paper | Discussion Paper | Discussion Paper | Discussion Paper |

**HESSD**

doi:10.5194/hess-2015-481

**Cclimate change and land use development effects on dam reliability**

P. Taherei Ghazvinei et al.

# 1 Introduction

Dams are integral elements in water resource with various purposes and beneficial roles. These include water supply, irrigation, flood mitigation and hydropower generation. Inappropriate design, operation and maintenance could have disastrous impact
on communities in the downstream area (Goodarzi et al., 2013). A statistical analysis on 534 dam failures from 43 countries revealed that the most common failure was associated with earth fill dam whereby 9 % of earth dam failures result from overtopping (Rico et al., 2008). As the importance of safety increases, dams have been viewed as a latent hazard in both developed and developing countries. It is necessary to ensure
dams operate safely throughout their life cycle (Hartford and Baecher, 2004). Failures of the dam structures by overtopping highlight the need of accurate assessment of their safety features such that emergency actions can be planned and implemented ahead of the probable catastrophic events (Yun et al., 2010).

In the study of flood risk management, assessment of possible dam failure is a major issue. Maximum flood for a structure to retain water before overtopping is called the design flood. Constructing a dam across a river effectively stores up enormous volume of water that could easily reach millions of cubic meters (Zhang et al., 2009). It is therefore crucial to ensure the dam structures withstand all possible forces that may be exerted on the structures and operate safely. Consequently, safety of all existing
dams needs to be re-assessed to determine whether the soundness of structure is acceptable for current standards (Blazkova and Beven, 2004).

Spillway is the common structure to divert the excess water and prevent a dam to have its design flood breached. If a spillway is undersized, the risk for a dam to get overtopped increases (Hartford and Baecher, 2004). The return period flood is determined
according to the probability of occurrence of extreme storm. The probable maximum flood, PMF is derived from probable maximum precipitation, PMP. The body of a dam is designed to adopt the probable maximum flood, also known as the spillway design flood (SDF) (Zhang and Singh, 2006).

Normally, historical climate data, coupled with the targeted performance, is used in the design of infrastructure systems and components (Taherei Ghazvinei et al., 2015). PMF usually varies according to the change of land use and climate. Change of land use, which affects the evapo-transpiration regime, has a direct effect on hydrologic pro-

cesses. On the other hand, the initiation of surface runoff is affected by the degree and type of ground cover (Fohrer et al., 2001; Tong et al., 2014). Changes in land use, such as deforestation and reclamation of wetlands, are common in developing countries as the rapid increasing population in many rural areas leads to need of more land for agricultural purpose (De Roo et al., 2001). This has negative impact on the surrounding

ecosystems and causes land degradation (Schreider et al., 2002). The river systems in many different worldwide regions have undergone changes in land use. Consequently, the storm water runoff generation and flooding regimes in these regions have been altered (Amini et al., 2014). Two processes can be distinguished in this context – the runoff generation within a catchment area and flow in a river network. Activities like

agricultural practice and urbanization influence the runoff generation. River engineering activities along river systems influence the discharge in a river network. A land mismanagement can have severe impacts on the hydrological regime which results in higher occurrence of flood and decline in dry season flows (Hundecha and Bárdossy, 2004).

It is evident that the climate is changing and this will continue. This makes the climate design ranges for a selected location unsuitable as reference (Ruth, 2006). Precipitation patterns shifted and bring more rain at certain location, while causing the other parts drier (Bates et al., 2008). For locations that are originally subjected to predictable precipitation, a change in runoff and recharging of groundwater is inevitable (Walther

et al., 2002). Temperature rise from the climate change also affect the evaporation of surface water and transpiration of vegetation, and leads to changes in the timing and frequency of precipitation (Wentz et al., 2007). All these changes pose threats to engineered systems which were not initially designed to the new precipitation patterns (Ruth and Kirshen, 2006). It is extremely difficult to take all possible changes into design

**HESSD**

doi:10.5194/hess-2015-481

**Cclimate change and land use development effects on dam reliability**

P. Taherei Ghazvinei et al.

Title Page

Abstract    Introduction

Conclusions    References

Tables    Figures

◁    ▷

consideration in the flood mitigation planning which involves a number of uncertainties. Improper selection of uncertainties will result in over or under design of a system (Hallegatte, 2009; Revi, 2008). Accordingly, climate change precipitation factor is needed in the design procedure, especially for updating. It intensity-duration-frequency (IDF)

curve (Jakob et al., 2011; Maraun et al., 2010) has been recommended to update the IDF curves based on Regional Climate Model (RCM) and Regional Hydroclimate Model (RegHCM) in the design procedure (Verma et al., 2010; Yener et al., 2007).

There is immense concern among scientists that improved knowledge and quantitative documentation on impacts of land use and climate change on dam safety is

needed. The regulatory authorities are now required to consider risk assessment on overtopping of dam as it may bring catastrophe to the downstream inhabitants.

The current paper analyses and discusses both empirical observations and modelling approaches related to impact of changes of land use and climate on flooding. The effect of climate change on flood is first summarized by presenting regional changed

precipitation, followed by discussion on hydrological and hydraulic processes. The discussion on modelling approaches includes overview of requirements of climatological and hydrological model for a case study. The final section presents summary on potential use of physically based hydrological models for analysing impacts from changes of the environmental conditions on the safety of dam.

## 2   Description of study area

Red Hill dam was constructed at 1906 to supply domestic water and mitigate the flood. The dam in constructed at downstream catchment with an area of 480 km$^2$ in the west region of the Peninsular Malaysia. Dam reservoir is located at the headwater region of Red, Kurau, and Fig rivers of its major tributaries. Red Hill dam is an earthfill embank-

ment dam with a height of 11.28 m equipped with two spillways.

Red Hill dam has an earth fill embankment while, upon completion, the embankment level was at RL 8.08 m above mean sea level, and subsequently improved to

**HESSD**

doi:10.5194/hess-2015-481

**Cclimate change and land use development effects on dam reliability**

P. Taherei Ghazvinei et al.

RL 10.67 m in 1965. Then in 1984 due to the Development Project of the Integrated Agricultural, level of the embankment was raised once more to the level of 11.28 m. At the current level, water is stored at 8.5 m RL to raise the water volume of the double cropping planting intensity to downstream irrigation area.

At the embankment crest level (ECL), the length of the crest is approximately 550 m and +11.28 m above mean sea level (MAMSL). The reservoir has the storage volume at 8.5 MAMSL level is 97.8 Mm$^3$ where, the lake surface area is 25 km$^2$. The design outlet capacity of the spillways at 8.5 MAMSL level is 210.4 cumecs. The physical characteristics of the dam catchments was obtained using the GIS tool from Digital Elevation Map (DEM). The area, longest flow path, centroid length, and average slope of the catchments was 486.5 km$^2$, 32.9 and 18.4 km, and 2.25 %, respectively.

## 3   Method of assessment

Evaluation process of the hydrological dam safety predicts dam overtopping possibility and the spillway adequacy. The tasks comprised of the PMPs' derivation at the project site, transposing PMPs to PMFs/SDFs by a function model of response or a catchment rainfall–runoff, and the flood rise estimation over full supply level (FSL) of the dam using a conventional technique of reservoir routing. In the present study, data of observed rainfall of 6 stations in Red Hill catchment were used to derive PMPs. The fundamental PMP convention was properly reviewed and assumed to be appropriate. Catchment response and Clark's Unit Hydrograph method was used to convert PMPs to PMFs for different periods (Links, 2016). Derived PMFs were routed through a reservoir using Hydrologic Modelling System (HEC-HMS). Hydrologic model was conducted using HEC-HMS to transform PMF to runoff (Verma et al., 2010).

**HESSD**

doi:10.5194/hess-2015-481

**Cclimate change and land use development effects on dam reliability**

P. Taherei Ghazvinei et al.

## 3.1 Data consistency

Rainfall analysing as a major input in hydrologic studies is required to establish PMP. The 5 min recorded rainfall data of auto-logger stations besides daily rainfall data of non-autologger stations were compiled and tabulated as the hydro-meteorological data. The rainfall pattern of Red Hill Dam was represented by 6 rainfall stations, which was named as Station no. 01 to 06.

Double mass curve analysis was conducted to quality control and consistency of the rainfall data collected from each hydrological station. Data consistency was checked based on the R square of the trendline through the plotted points of cumulative rainfall data for each rainfall station against the cumulative rainfall data obtained from the other nearby rainfall stations. Considering the double mass rainfall consistency test, Thiessen Polygons method was used to interpolate rainfall records of the stations that are located in and on the border of the Catchment. Thiessen method makes allowance for the areal distribution of the rainfall station. After data correction, double mass curves were again plotted to check whether the adjusted data was significantly consistent. Figure 2a is a representative of the rainfall data double mass tests. By achieving significant consistency of the rainfall data based on the double mass curve, isohyets curves of the highest recorded rainfall for 1, 3, 5, and 7 day durations was drawn. These isohyets map provides a general understanding of the rainfall distribution in the area. Figures 2b presents the isohyets map for 1 day duration.

## 3.2 Updating rainfall intensity duration frequency (IDF) using RCM

The IDF curves were developed from the raw rainfall data collected from the hydrological stations (auto-logger) located in the Red Hill catchment. The annual maximum rainfall of different durations have been sorted out from these stations and analysed by adopting the Gumble Distribution Method.

Design storms had been produced in Urban Stormwater Management Manual for Malaysia, (MSMA, 2012) using rainfall data up to the year of 2010 for selected rainfall

Discussion Paper | Discussion Paper | Discussion Paper | Discussion Paper

**HESSD**

doi:10.5194/hess-2015-481

**Cclimate change and land use development effects on dam reliability**

P. Taherei Ghazvinei et al.

station in Malaysia as well as for Chart station which is the nearest rainfall station to the Red Hill catchment area (see Fig. 1). In the current study, the rainfall intensity analysis was carried out using the recent 30 years rainfall data up to year of 2015 of the Chart station. A comparison between updated rainfall intensity and the last published, indicated that the intensity of rainfall for different duration has been increased. Figure 3a shows the results of comparison for 100 year return period Average Recurrence Interval (ARI). Therefore, any rainfall analysis should include the latest rainfall data from 2011 to 2015, especially for calculating design storm.

Time and runoff of the maximum discharge of the flood peak are affected by the rainfall temporal distribution which is considered to be a vital factor within the design storm. Design rainfall temporal pattern of actual storm pattern for Region 3 (including the Red Hill area) was adopted from Hydrological Procedure No. 1 (Estimation of Design Rainstorm in Peninsular Malaysia) (DID, 2010) to compare with the typical change in intensities of the rainfall during a usual storm burst.

## 3.3 Climate change impact

The expected impacts of climate change on water resources, which can be valued in terms of physical level or change in flood magnitudes and consequently could be translated into the cost for adaptation in designing and developing water infrastructure systems and projects. For future rainfall intensity the equation is shown below:

$$I_{Future} = I \cdot \text{Climate Change Factor (CCF)} \tag{1}$$

Where;

$I$ = rainfall intensity (mm hour$^{-1}$)

The Climate Change Factor (CCF) for Peninsular Malaysia for different duration and locations were produced by National Hydraulic Research Institute of Malaysia (NAHRIM, 2013) where CCF for Peninsular Malaysia is proposed from 1.03 to 1.28 for 2 to

Discussion Paper | Discussion Paper | Discussion Paper | Discussion Paper |

**HESSD**

doi:10.5194/hess-2015-481

**Cclimate change and land use development effects on dam reliability**

P. Taherei Ghazvinei et al.

100 years ARI, respectively. Using CCF, the IDF curves for the stations at catchment area including CCF were updated. Figure 3b shows the IDF curves for the station with the highest rainfall intensity at catchment area including CCF.

## 3.4 The impact of development plan on landuse

Bronstert et al. (2002) stated that the generation of storm runoff is influenced by the practices of land-use to the near-surface or surface of the soil at least under normal conditions while, land-uses at subsurface soil zone such as mining are not considered. Therefore only the near-surface or surface storages and fluxes are affected by land-use changes.

Detailed land use activities was identified from the Development Plan established by District Council and the Department of Town and Country Planning. In general, the land use in the study area is essentially composed of three main land use types:

i. Forest and water body mostly occupied in upper sub-catchment that comprises approximately one-quarter of the basin area;

ii. Agriculture is a significant sector as it comprises approximately more than half of the total land utilization basin;

iii. Built-up area and aquaculture land use occupies a small percentage of the total river basin area.

Results showed that the imperviousness of the area will be increased up to 4.5 % by
the year of 2030. Figure 3c shows the summary of the results of land use for future development condition of the study area compared with the year of 2015 condition. The impacts of the imperviousness change on the direct runoff is predicted by the hydrological model.

Discussion Paper | Discussion Paper | Discussion Paper | Discussion Paper

**HESSD**

doi:10.5194/hess-2015-481

**Cclimate change and land use development effects on dam reliability**

P. Taherei Ghazvinei et al.

### 3.5   Probable maximum precipitation (PMP)

Probable Maximum Precipitation (PMP) indicates the precipitation/rainfall upper limit under apparent and favourable participating factors, such as availability of moisture, and moisture barrier presence/absence at higher mountain range in the storm move-
ment track. In this study, following methods were adopted to estimate PMP:

#### 3.5.1   The statistical method

The maximum annual rainfalls for the 1 to 7 days duration were abstracted from each station. Stations' PMP were predicted using Hershfield (1965) equation.

#### 3.5.2   The envelope curve

Hershfield (1965) showed that the estimation of PMP is independent of the single highest value of frequency factor $K_m$. Therefore, he empirically established various $K_m$ that is inversely related to the mean of the series and direct relation with rainfall duration. In other words, as the mean of the series is increased in magnitude, $K_m$ value has a tendency to decrease. Therefore, each station has its own $K_m$ value based on its mean
magnitude. Subsequently Hershfield (1965) used the envelope $K_m$ curve to calculate $K_m$ values of the mean of the series. As such, appropriate envelope curves equations for different durations were applied to calculate $K_m$, corresponding to the values of mean annual maximum rainfall (NAHRIM, 2008). Next, The PMPs of each station was estimated using the Hershfield equation.

#### 3.5.3   The storm maximization and transposition

This method assumes that the PMP will result from a storm in which there is the optimum combination of the available moisture in the atmosphere and the efficiency of the storm mechanism. The observed rainfall from a historic storm is used as the indirect measure to estimate the storm efficiency. Storm maximization and transposition

Discussion Paper | Discussion Paper | Discussion Paper | Discussion Paper

**HESSD**

doi:10.5194/hess-2015-481

**Cclimate change and land use development effects on dam reliability**

P. Taherei Ghazvinei et al.

techniques are incorporated for estimation of PMP for the targeted area in order to compensate for the deficiency of a sufficient storm database. The details of analyses are available in Casas et al. (2008).

Review of rainfall data in the List of Hydrology Data (DID, 2015) indicated that Pintu Kawalan storm in Johor, in southern part of Malaysia has the most historical severe storms in the country for November 1980, where the 1 day and 3 days rain had recorded rainfalls of 924 and 1700.5 mm. Pintu Kawalan storm was transposed into the study area to calculate the PMP for 1 h to 3 days (Table 1).

## 3.6 PMP/PMF routing

PMF indicates a flood that may result from the strictest combination of critical hydrologic and meteorological conditions defined by Swain et al. (1998). PMF also has physical meanings which provide an upper limit of the interval, an engineer must operate and design. Today, PMF is commonly recognised as the standard for the dam safety design where the incremental penalties of failure have been identified to be intolerable (Kuo et al., 2008). Therefore, prediction of extreme floods is vital in dam safety and hydrologic engineering (Swain et al., 2004). Extreme flood mathematical watershed models can be used in simulation where, mostly the storage routing or unit hydrograph models are widely used in watershed models for the simulation of extreme floods and PMF.

Routing PMF on the dam reservoir is used to determine whether the dam spillway capacity is adequate to avoid dam crest overtopping. For estimation of the incoming flood into a reservoir, a suitable procedure is required to convert the PMP into PMF. The process of the conversion of PMF using PMP is typically carried out using a conventional rainfall runoff routing by convoluting the produced runoff on the basis of rainfall temporal distribution. Converting and routing by convolution of temporally distributed PMPs into PMFs of different rainstorm periods; i.e., from 1 to 120 h form one of the vital tasks in a typical PMP/PMF study. The most commonly used techniques beyond numerous existing hydrological rainfall runoff procedures in the local context are, (1)

**HESSD**

doi:10.5194/hess-2015-481

**Cclimate change and land use development effects on dam reliability**

P. Taherei Ghazvinei et al.

Discussion Paper | Discussion Paper | Discussion Paper | Discussion Paper

hydrological procedure No. 11 (HP 11, 1994) on flood estimation and (2) modeling approach using proprietary as well as non-proprietary mathematical models/software hydrological procedure No. 27 (HP 27, 2010). HP 11 presents a deterministic method of estimating the design flood hydrograph for ungauged rural catchments in Peninsular Malaysia. The procedure is based on the development of three components: a design storm, a rainfall–runoff relationship, and a triangular hydrograph. The procedure was tested on 12 gauged catchments and gave average results while, in HP 27 the method of the design flood hydrographs estimation for rural catchments in Peninsular Malaysia has been presented also by the equations for Clark parameters in the development of design flood hydrographs.

Generally, the conversion of PMP to PMF needs the approximation of retention dam-ages, the convolution of excess rainfall and unit hydrograph, the derivation of unit hy-drographs, the flood routings through the reservoir and the choice of antecedent and consequent floods.

In this study, the deterministic approach was chosen in determining PMF for the catchment. The HEC-HMS software developed by US Army Corps of Engineers Hy-drologic Engineering Center was chosen for this study. It contains many of the familiar and well applicable hydrologic methods to be used for the simulation of rainfall–runoff procedures in river basins (Chu and Steinman, 2009). This method attempts to char-acterise the strictest combination of critical hydrologic and meteorological conditions considered rationally possible for certain drainage basin.

PMP was used as the meteorological input. Basin physical parameters that include area of sub-catchments, stream length, and imperviousness area were obtained us-ing Arc-GIS. The basin model was developed in HEC-HMS (Fig. 4), which consists of the hydrologic elements such as sub-basin, reach, junction and reservoir representing a physical process such as catchment, stream reach and confluence.

Initial and Constant Rate method was used to compute the runoff volume for use in the determination of catchment PMF values, where the maximum potential rate of precipitation loss is assumed to be constant throughout an event. Clark's Unit Hydro-

**HESSD**

doi:10.5194/hess-2015-481

**Cclimate change and land use development effects on dam reliability**

P. Taherei Ghazvinei et al.

graph method was used to model the direct runoff based on the constant monthly base-flow method. The HEC-1 Clark unit hydrograph requires the specification of the watershed/basin drainage area A, the time of concentration Tc, the linear reservoir's storage constant K, and the time-area histogram. In addition, Lag method is used to model the routing. As a next procedure, unit hydrograph was produced to convert the direct runoff into flood discharge hydrograph.

The principal objective of the hydrological modelling was to simulate flood runoff for Red Hill Dam based on the historical rainfall recorded, climate change effect, and development plan of the area which serve as input for the hydrological model. The HEC-HMS model needed the input data of time-series data with discharge and precipitation of particular periods for calibrations and relationship of the spillway elevation-discharge and dam reservoir elevation-storage. In model calibration, severe historical rainfall used was considered. The daily discharge raw data for the flow stream station no. 1 located upstream of reservoir was authenticated by plotting. Subsequently, two events were selected as follows: 8–19 June 2004 for model calibration, and 18–29 December 2007 for the model verification.

The optimization manager in HEC-HMS model was used to evaluate fitted parameters for model calibration based on variance between computed and observed discharges. For model verification, the typical parameters of two model calibrations were used.

## 3.7 Spillway adequacy

For reservoir parameters, the elevation-storage outflow method was chosen for this study. The elevation-storage curve and the elevation-discharge curve for the dam were plotted based on the dam structure design and GIS topography analysis of the area. Reservoir elements for the dam consist of one intake, two spillways and a saddle dam.

The outlet structures of a dam such as spillways and sluice gates must be capable to protect the main dam body through the evacuation of a dangerous flood of PMF magnitude. The storage change rate in the reservoir water body is the quantification and

**HESSD**

doi:10.5194/hess-2015-481

**Cclimate change and land use development effects on dam reliability**

P. Taherei Ghazvinei et al.

summation of all inflows from different sources and properly subtracting the outflow amount through outlet structures, for example bottom outlets or spillways of a reservoir/dam. The bottom outlet flow and other losses for instant seepage through the dam body are assumed to be negligible for simplification. At the start of a PMF event the reservoir is expected at its full supply level.

## 4   Results and discussion

The results of this study are given in two parts, the derivation of PMP/PMF using an up to standard catchment routing method based on Clark's hydrograph and on the outcomes of reservoir routing and flood rise estimation using a modified approach accredited to HEC-HMS.

The point PMP values at the catchment stations as well as transposed from the severe historical storm were calculated (Fig. 5). The observation showed that the PMP derived statistically is not equivalent to the PMP estimated using the Hydro-meteorological method. The reason is that the statistical method applies extrapolation of frequency distribution fitted to its historical data series of the annual maximum rainfall values which has limited length and questionable quality. The extrapolation of short lengths of rainfall records is unreliable due to the randomness in space and time during the occurrence of major rainstorms. Whereas, Hydro-meteorological method uses meteorological reasoning to develop the PMP for different durations and the derived PMP is the outcome from a storm in which there is the optimum arrangement of the obtainable moisture in the atmosphere and the effectiveness of the storm mechanism. The PMP derived from the Hydro-meteorological method is more than the PMP estimated from the statistical method.

PMP values derived using statistical approach is significantly underestimated when compared to the design rainfall storm. Thus, the PMF derived through the above PMP will not be favourable for critical flood simulation for dam safety analysis. PMF based on the PMP of Hydro-meteorological method have resulted in an extreme flood with

**HESSD**

doi:10.5194/hess-2015-481

**Cclimate change and land use development effects on dam reliability**

P. Taherei Ghazvinei et al.

**HESSD**

doi:10.5194/hess-2015-481

**Cclimate change and land use development effects on dam reliability**

P. Taherei Ghazvinei et al.

an extra high risk for dam safety. Therefore, to route a floods the PMF derived from PMP that uses statistical envelope curves including CCF were selected to simulate the flood runoff on the dam because corresponding PMP value of 1 day was consistent with PMP value of 24 h in the ARI of 100 years derived from regional PMP of design rainfall (Fig. 3b).

## 4.1 PMP/PMF Catchment Routing

In this study, PMFs were derived from PMPs of 1 h up to 72 h storm durations. Subsequently, entire temporal distribution of PMP was used to integrate conventional convolution. The hydrograph of simulated flood based on the PMF for different storm durations are plotted as shown Figs. 6. The PMF (from 12 h PMP) hydrograph generated represents the maximum storm with the highest peak discharge compared to other storm durations.

## 4.2 Reservoir Routing

The flood hydrographs were routed through the reservoir. According to the operation records of the Red Hill Dam, data from 2000 to 2014 shows that normal frequency water level is 8.5 MSML and the release level of water is 9 MSML. Therefore, initial reservoir level was assumed at EL 8.5 MSML. The data inputs used in the flood routing analysis are reservoir level-storage curve and flood hydrograph for 50 years ARI, 100 years ARI, and PMF with a 1 h, up to 72 h storm duration. The reservoir routing procedure has also considered the worst case condition where the outlet works fail to operate. The curve of the total discharge capacity of the spillways (with maximum of 589 cumecs) is presented in Fig. 7a. Analysis has found that the maximum outflow discharge could have occurred upon opening of the spillway gates as storage elevation was noted above the spillway crest elevation.

Flood routing simulation showed that the maximum flood level derived from the critical duration of PMF is under the embankment crest elevation of the dam. However, to

ensure dam is in a safe condition, the freeboard should not be reduced to less than 1.00 m to avoid overtopping at the dam crest. Flood on the dam was routed for current and future development condition. For the routing procedure, there are two possible return periods (ARI) including the CCF (Climate Change Effect) and PMF translated from PMP derived using Statistical Hershfield (Curve) Method with a 1 h, up to 72 h storm duration in two scenarios:

**HESSD**

doi:10.5194/hess-2015-481

**Cclimate change and land use development effects on dam reliability**

P. Taherei Ghazvinei et al.

Discussion Paper | Discussion Paper | Discussion Paper | Discussion Paper |

**HESSD**

doi:10.5194/hess-2015-481

**Cclimate change and land use development effects on dam reliability**

P. Taherei Ghazvinei et al.

i. Scenario I:

- Case 1: 50 years ARI Design Flood,
- Case 2: 100 years ARI Design Flood,

ii. Scenario II: PMF translated from PMP derived using Statistical Hershfield Method.

Table 2 summarises the routings for future development and the current conditions. It was predicted that discharge of the direct runoff with 50 ARI for the year 2020 will increase by 2.29 % from the current condition. It can be observed that the highest peak inflow occurred during the 12 hour storm duration of PMF while the most critical flood within the least free board is due to flood based on PMF of 72 h. Inflow, outflow, and changes of the reservoir elevation curves of the most critical flood are shown in Fig. 7b where, "Combined Flow" represents "Inflow". The right vertical axis shows the elevation condition for the pool and flood elevation where the left axis shows storage volume and flow discharge for upper and lower graphs, respectively.

Analysis on the reservoir routing results show that spillway of the dam is capable to operate under the design flood conditions (up to 100 years ARI) and unlikely be faced with overtopping problems. In safe operation condition, the potential storage behind the dam has sufficient supply for design flood inflow. In addition, it is revealed that Red Hill Dam is marginally safe for the inflow of PMF with the duration of 24 and 72 h if the free board is decreased to 1.00 m.

## 5 Conclusions

A hydrological dam safety assessment was conducted with the objective to check the adequacy of the dam spillways in view of an excessive meteorological happening of the PMF magnitude considering the effects of climate change and land use development. Red Hill Dam selected for this study has dual functions to alleviate flood and as reservoir for water supply. The study adopts 3 types of PMPs based on the close vicinity of

the catchment upstream of the dam and the rainfall stations, while the maximum rainfall records were adopted in the PMP derivation. A method of catchment routing was used to interpret the PMPs to PMFs for 1 to 7 day periods. Two scenarios are performed; namely: (1) 50 and 100 years ARI Design Flood, (2) PMF translated from PMP derived using Statistical Hershfield Method. The maximum flood levels at the dam reservoir were found to be lower than the crest level of the dam embankment. For such a flood an adequate capacity of the dam reservoir is available to store the flood, simultaneously the water rises up to the level of 9.9 MSL where the dam is safe. In other words, for the case with the free board close to 1.0 m, the spillways of the dam marginally have adequate capacity to safely pass the extreme flood up to PMF for the year of 2020. Although, the dam is marginally safe, there is a requirement to alter hydraulic control and to improve the spillway adequacy (special structural and construction plan), using flood forecasting system.

*Acknowledgement.* The authors express their sincere thanks for the support of UM.C/HIR/MOHE/ENG/47 and UM.GC002A-15SUS.

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

**Table 1.** PMP for the study area by hydrometecrological method (storm transposition).

| PMP at the Dam (mm) | | | | | |
|---|---|---|---|---|---|
| 1 h | 3 h | 6 h | 12 h | 1 day | 3 day |
| 518.8 | 679.1 | 697.4 | 711.9 | 1298 | 2356 |

Discussion Paper | Discussion Paper | Discussion Paper | Discussion Paper | Discussion Paper |

HESSD

doi:10.5194/hess-2015-481

**Cclimate change and land use development effects on dam reliability**

P. Taherei Ghazvinei et al.

**Table 2.** Comparison of peak inflow between Current Development Condition and Future Development Condition (2030) for maximum design floods and PMF.

| Maximum design flood with × hour duration and year ARI, and PMF derived from 12 h PMP | Time of the peak discharge, $T_p$ (hour) | | $q_p$ (cumecs) | | $q_p$ difference % |
|---|---|---|---|---|---|
| | Current | Future | Current | Future | |
| 50 yr ARI (3 h) | 7.33 | 7.33 | 542.1 | 554.5 | 2.29 |
| 100 yr ARI (3 h) | 7.33 | 7.33 | 594.7 | 607.1 | 2.06 |
| PMF (12 h) | 10.83 | 10.83 | 1664.5 | 1679 | 0.87 |

Discussion Paper | Discussion Paper | Discussion Paper | Discussion Paper | Discussion Paper |

**HESSD**

doi:10.5194/hess-2015-481

**Cclimate change and land use development effects on dam reliability**

P. Taherei Ghazvinei et al.

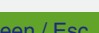
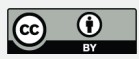

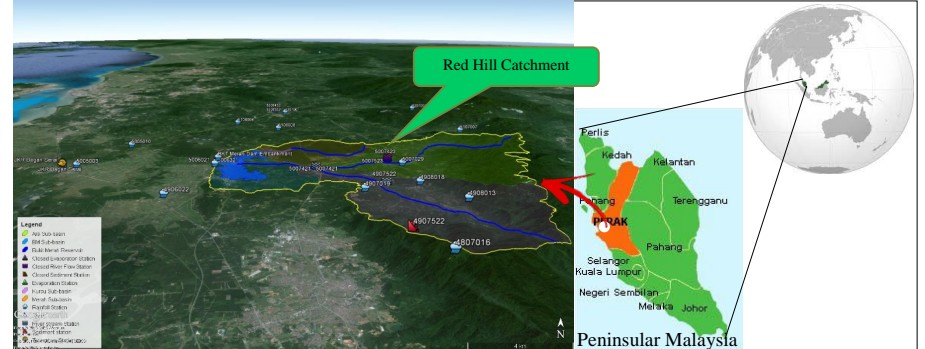

**Figure 1.** Location of Red Hill Dam and the borders of the upstream catchment.

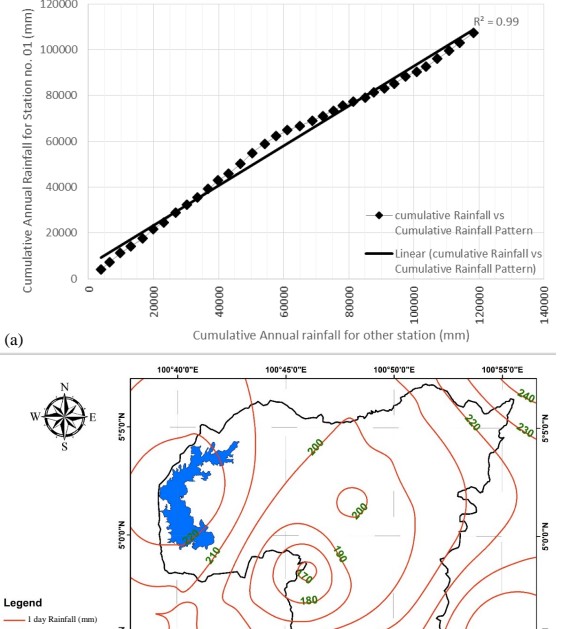

(a)

(b)

**Figure 2. (a)** Double mass curve for rainfall data of station no. 01 in the catchment, **(b)** Isohyets map using the highest recorded rainfall values in mm for 1 day duration over the area.

**HESSD**

doi:10.5194/hess-2015-481

**Cclimate change and land use development effects on dam reliability**

P. Taherei Ghazvinei et al.



Discussion Paper | Discussion Paper | Discussion Paper | Discussion Paper

**HESSD**

doi:10.5194/hess-2015-481

**Cclimate change and land use development effects on dam reliability**

P. Taherei Ghazvinei et al.


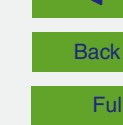
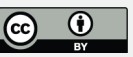

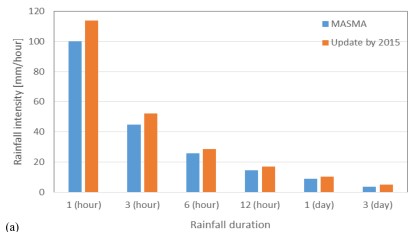

(a)

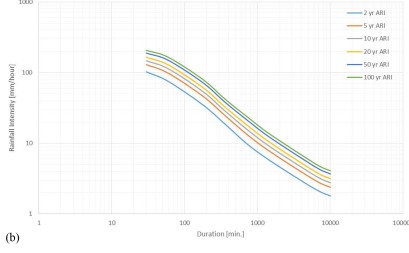

(b)

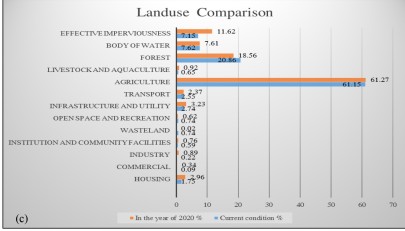

(c)

**Figure 3. (a)** Comparison of rainfall intensity (100 ARI) MSMA 2nd edition, 2012 (based on HP1) and updated data (2015) of chart station, **(b)** updated IDF curves with climate change factors for stations with highest rainfall intensities, **(c)** land use percentage in the study area for current condition and at the year of 2020.

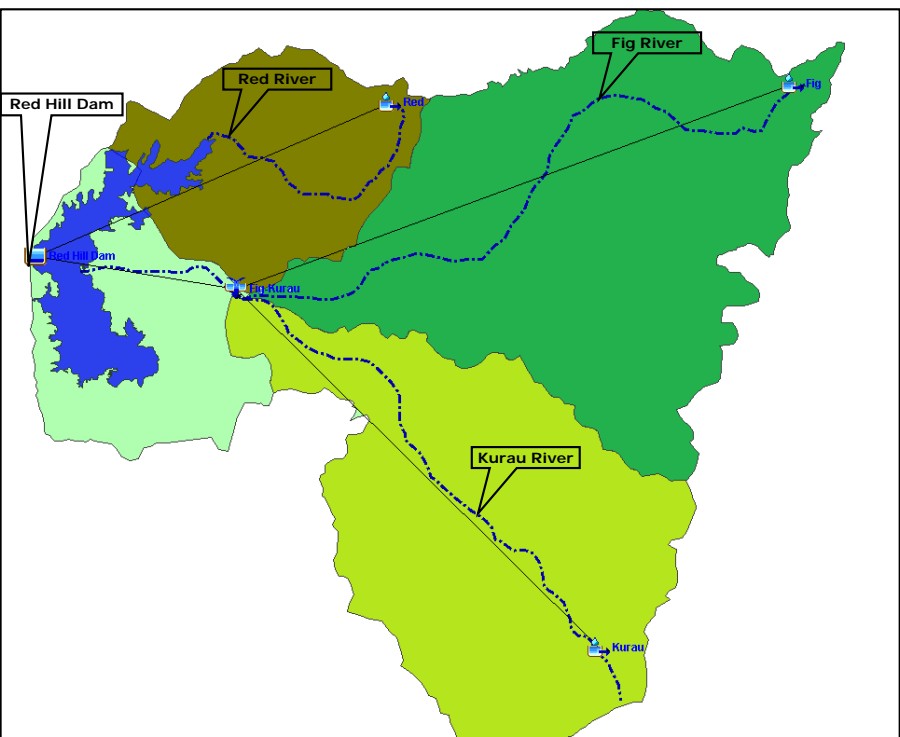

**Figure 4.** Basin model of the catchment.

Discussion Paper | Discussion Paper | Discussion Paper | Discussion Paper | Discussion Paper |

# HESSD

doi:10.5194/hess-2015-481

**Cclimate change and land use development effects on dam reliability**

P. Taherei Ghazvinei et al.

Discussion Paper | Discussion Paper | Discussion Paper | Discussion Paper

**HESSD**

doi:10.5194/hess-2015-481

**Cclimate change and land use development effects on dam reliability**

P. Taherei Ghazvinei et al.

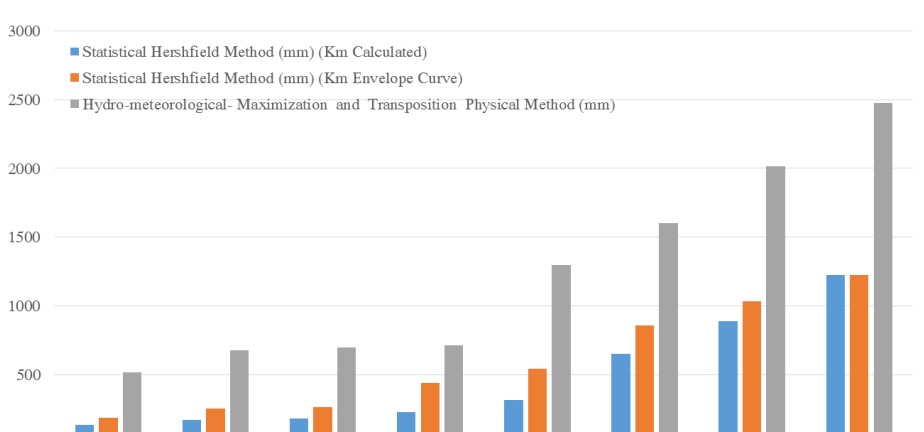

**Figure 5.** The PMP values of study area using three approaches.

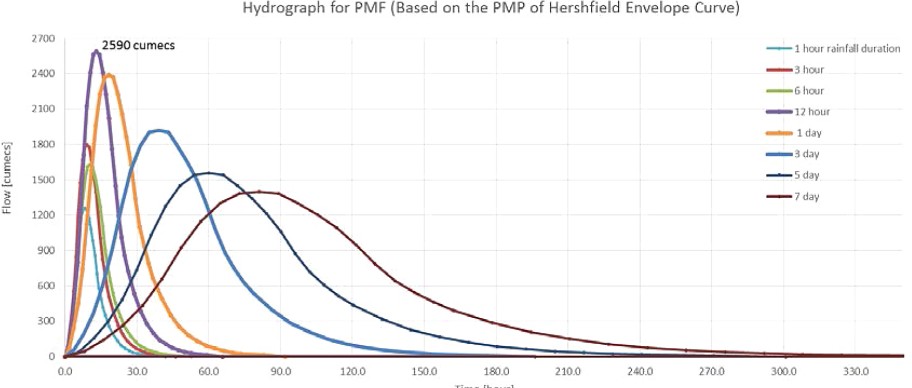

**Figure 6.** Hydrograph of the PMF using Hershfield envelope curve method for 1, 3, 6, 12, 24 and 72 h durations.

**HESSD**

doi:10.5194/hess-2015-481

**Cclimate change and land use development effects on dam reliability**

P. Taherei Ghazvinei et al.



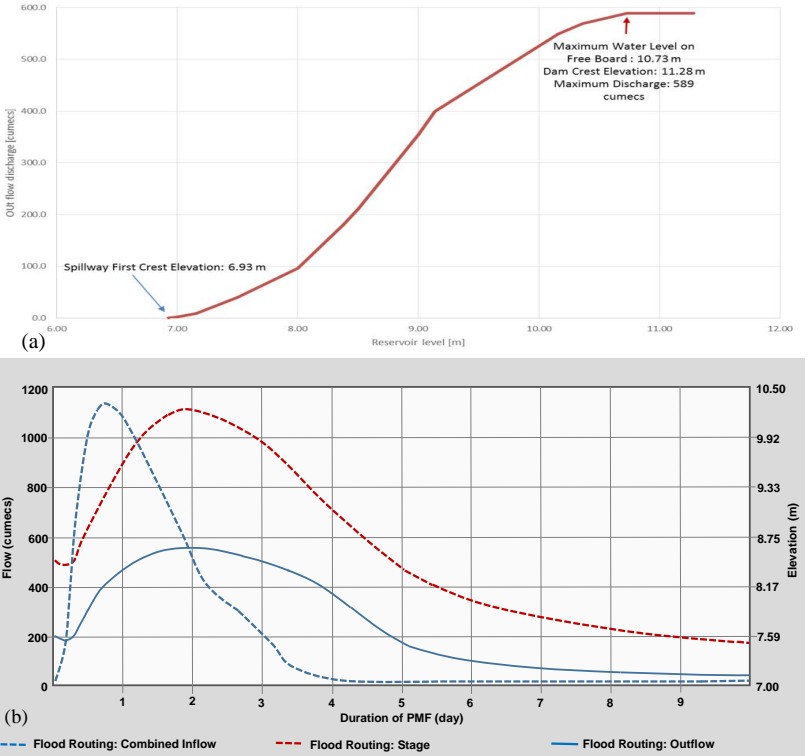

**Figure 7. (a)** The curve of the total discharge of Red Hill Dam spillway, **(b)** Inflow-outflow hydrograph of the most critical flood of the PMF with duration of 72 h.

**HESSD**

doi:10.5194/hess-2015-481

**Cclimate change and land use development effects on dam reliability**

P. Taherei Ghazvinei et al.

Discussion Paper | Discussion Paper | Discussion Paper | Discussion Paper