# Peer review of "Assessment of climate change and land use development effects on dam reliability"

_Hydrology and Earth System Sciences, 2015_

## Referee Comment (RC1) · Anonymous Referee #1 · 25 Jan 2016

The Manuscript titled "Assessment of climate change and land use development effects on dam reliability", presents the findings on a particular case study of the effect of climate change and land use on Dam safety. The study is interesting and worth publishing. Nonetheless the title should be changed and there are several questions and remarks that I would request the Authors to address.

I have divided my review into major issues and minor comments:

Major issues:

The first issue is to do with the title of the paper. The title implies that dam reliability is assessed. (Please see Reliability index references, e.g. http://ascelibrary.org/doi/abs/10.1061/%28ASCE%29GT.1943-5606.0000313) However I cannot find it in the text. My suggestion is that the authors rephrase their

title and omit the use of Reliability on their Manuscript and title.

My second issue is to do with the climate change analysis. I would request the authors to show the rainfall data of the last 30 years which was used to produce figure 3. With this data the authors should do a trend analysis tests to support their claim that intensity is indeed increasing. The comparison with MSMA is not convincing. Unless the authors could justify that Gumble distribution was also used in the MSMA study. In any case it would make a better statement if the authors could add a data trend analysis to their work.

Still related to my second issue, I would request the authors to add a discussion on the required length of a data series, to capture a climate change trend. This discussion should be added to the introduction. The Authors could than criticize if 30 years might would be enough to fit their aim, i.e. of the trend for extreme events (100 return period).

My third concern, is related with the hydrological modelling. In my view, there is a section missing about calibration and validation of the hydrological model. The hydrological modelling was done based on historical rainfall recorded (line 12, page 14), however I do not see any results. The authors show add the results of their calibration/Validation strategy, and perhaps add some measure of fit to show how good the model performed.

Minor comments

Please add a small profile of the spillway, with dimensions of the spillway crest level, embankment crest level, etc,. . .

Please add some details of the Hershfield statistical method (including envelope), and the transposition method.

What is the spatial distance between the case study and the one where the storm occurred?

Introduce all acronyms in the abstract

[Figure]

Besides the abstract and introduction, the manuscript needs to be re-written in many places. At some points it is difficult to understand the meaning of some sentences. Please find below a few examples, but more can be found throughout the text.

Line 4 -8,page 6:: sentence is unclear.

Line 27,page 6: introduce "RL" acronym.

Line 15,page 7: which one was used: "by a function model of response or a catchment rainfall–runo_,"

line 22 -23, page 7, sentence is unclear.

line 1 -2, page 8, sentence is unclear.

Line 7-8, page 8,the sentence is missing a verb? maybe check?

Line 6, page 8, replace "was" with "were"

Line 13, page 8,the sentence is unclear

Line 10, page 10, replace "was" with "were"

And many others. . .

―――――――――――――――――――

---

## Referee Comment (RC2) · Anonymous Referee #2 · 17 Feb 2016

Review

"*Assessment of climate change and land use development effects on dam reliability*" by P. Taherei Ghazvinei, H. Hassanpour Darvishi, R. B. Hashim, S. H. Musavi Jahromi, and N. Aghamohammadi

The manuscript under review presents a very applied study and claims to investigate how climate change and land use change in a catchment in Malaysia affects the dam reliability (which I would call safety) in some decades. Three variations of floods were simulated and it is concluded that those do only have a marginal effect on the water levels inside the reservoir/impoundment under investigation.

The topic generally addresses issues of importance to the readership of the journal. However, the manuscript is generally not well written, it lacks critical discussion and in most parts it is very difficult to read due to language issues. Based on the number of comments, questions, and concerns extended below, the manuscript in the present form is not yet mature enough to be considered suitable for the permanent literature.

Assessment: It is thus reasoned to not recommend it in its current form for publication.

Major and minor detailed comments are provided below to help the authors to further develop their manuscript.

General comments:

1.  The manuscript does not read well in technical and linguistic terms. Before publication, it is consequently necessary to have it proof-read by a native speaker. The reviewer started suggesting language and expression improvements for the abstract but stopped as it was too tedious to continue throughout the manuscript. This needs a professional service before being ready for publication.
2.  The introduction only partially touches the relevant literature. I cannot see any reference to the manifold bulletins which are published by the International Commission on Large Dams (ICOLD) which is a standard resource of information in regard to dams in general. It would suit the purpose of this manuscript to show that the findings (which are much on the applied side anyway) are well in agreement with those stipulations on dam constructions.
3.  The introduction of the PMF is somewhat weak and deserves a more thorough description in order to allow readers understand what is spoken about later on. E.g., it is not enough to reduce the PMF to be influenced to change of land use and climate. It definitely is also a function of the drainage area, its topography and slope characteristics, hence the whole interplay of factors needs at least to be mentioned. In this context it would be interesting to also compare the author's results of conversion of PMP to PMF against empirical envelope methods such as the Creager or the Francou-Rodier equation which relate the peak flow with the drainage area. Please direct a comparison towards these empirical relations and discuss how this fits into the climate change influence discussed herein.
4.  The introduction misses out on a key element in scientific papers: the level of novelty is not addressed at all. This observation aligns with the lack of clarity in terms of objectives and goals which are not found to be mentioned in the introduction. It is required to more clearly state what objectives the authors pursued and where those objectives are tied into the lack of knowledge which ideally was found from a proper

literature review/discussion. All those elements I cannot find easily in the introduction and thus it needs significant improvements.

5. Besides all the issues regarding the methodology section provided below, there is one main issue the reviewer holds against the authors. It relates to the modelling of the flow in the vicinity of the dam site which usually is composed of overflow conveyed over the spillway and the flow released through the sluice gate at the bottom of the dam. It does not become clear why there was no more modelling involved other than the parametric model HEC-HMS aiming at simulating the system on the catchment scale. In particular the flow through the spillway requires more sophisticated means of simulation, either 1D with e.g. HEC-RAS, or given the steep slopes of many spillways worldwide, better 2D or 3D models linked together in a model cascade.

6. It is also important to look into backwater effects which might occur downstream of the dam site effectively reducing the flow capacity of the spillway and river cross-section involved. The current work does not convince that there has been enough focus on those local effects which however become crucial when looking into dam reliability. And again, there is no information regarding the actual dam structure which makes it extremely difficult to judge what is going on there. The authors need to provide a fair bit of information to let the audience value the work by themselves.

7. The result and discussion section is particularly disappointing to read. From what is promised in the manuscript title one would expect to learn how the climate change and land use variation over the years would contribute to increase PMF used for the design of a dam as an application. But there is no more distinguishing between the two factors the authors set out to focus on. From the title, readers would expect to know how a single change, say of climate only, would affect the design for a dam. What is more, the manuscript suffers greatly from the unclear language in which it is written. There is a great need to be more specific and it is advised to use as many references as possible to underpin the author's case.

8. Section "Conclusion" should either be renamed to "Summary" or there needs to be actual meaningful conclusions to be drawn which will eventually help others to learn how to address climate and land use change with respect to dam reliability.

9. Moreover, it does not seem convincing that the maximum water level inside the impoundment just marginally increases and that the dam is still safe given the drastic increase in rainfall detailed in the methodology section. It might be more important to vary the climate change factor quite a bit more in order to see how these variations will affect the water levels inside the impoundment. Also, there is a need to look into the processes where the spillway is involved.

More detailed comments (keyed to page (P) and line (L) numbers):

1. P3/L9: Wrong/imprecise wording "Flood rises". Consider words like flow depth, surface elevation, flood level
2. P3/L9: Please write "Considering the climate change factor …"
3. P3/L11: Please revise to "…, the year of 2020…"
4. P3/L14: Remove "of" before 2030
5. P3/L16: Define PMP, PMF, HEC-HMS also in Abstract.
6. P3/L16: Please modify to: "The software HEC-RMS…"
7. P3/L17: "flood rises", please see above comment
8. P3/L17: Revise the double use of "and" in the sentence.

9. P3/L18: Mention what "modified technique was used" or remove from abstract.
10. P3/L20: Use "investigated" instead of performed.
11. P3/L23: What is "marginally adequate", reads awkward.
12. P3/L25: Please use: "…, the dam safety in terms of hydrology was assured."
13. P4/L8: "As the importance of safety increases…" Please focus this sentence more clearly, it reads very unspecific. Without changes in external forcing such as climate change, there would definitely be no reason to re-assess dam safety as dams are usually build according to well-drafted design codes which include measures for safe constructions even under extraordinary loads.
14. P4/L19ff: Please add the exact reason why a re-assessment is necessary. This does not become clear without consulting the reference.
15. P4/L22: "Spillways are a common way to…"
16. P4/L23: Consider reformulating: "If a spillway is not designed properly…"
17. P4/L26: Please enclose abbreviations in brackets.
18. P4/L 27: "The body of a dam…" The actual design criteria needs to be explained in more detail. In particular, it is not about the body of the dam, but about the amount of water that can be stored in the dam impoundment and conveyed through the overflow spillway such that no crucial part of the dam construction is affected. This needs to be detailed in a more technically correct way.
19. P5/L1: What is normal in the context of historical climate data? This statement needs revision. Also, the sentence requires more explanation on what the "targeted performance" is. It appears as if this is a portion of some other work the first author led, but it is difficult to understand without further explanation. E.g., it would be advantageous to learn, for which types of infrastructure these statements were made and whether the methodology is applicable to dams in general and earth filled dams in particular.
20. P5/L17: Please detail the term "mismanagement" in this context. The term includes a pre-judgement and needs additional justification.
21. P5/L20: Refrain from stating the obvious yet highly contested without properly referencing.
22. P5/L20f: Sentence difficult to understand, please revise.
23. P5, L21f: Incorrect grammar (tenses mixed).
24. P6/L4: Grammar issue "It …"
25. P6/L25: What height is referred to? Crest height? Please be specific. Is there a local datum to which this is referred to? Please state the important facts.
26. P6-Fig1: Figure is neither mentioned nor explained. Please tie in the Figure and introduce the study area to the readers.
27. P7/L1: What is "RL", please always write in full the abbreviation used.
28. P7/L1f: What is "Integrated Agricultural"? Unclear.
29. P7/L3f: "… double cropping planting …" Unclear what this sentence should say? Needs revisions.
30. P7/L8: Please always use SI units and its derivatives. I assume this is m^3/s?
31. P7/L9: "the GIS tool …" What GIS tools were used and what methods were applied to yield the information stated? Please be precise in your description of the work.
32. P6/7: Section 2 requires more visual details, e.g. provide a plan view of the dam and impoundment. Detailed overview over the spillway and related measures taken at the dam site would also greatly help the reader understand how the spillway is constructed as its cross-sectional area, slope, energy dissipation mechanisms will most certainly affect how well storm water discharge is conveyed downstream. This is

essential to this study which in my understanding exactly addresses this questions under climate change aspects.

33. P7/L19: "fundamental PMP convention". Please detail, unclear. Needs citation/reference.

34. P7/L22: Please properly cite the software the authors used.
    USACE, "Hydrologic modeling system," HEC-HMS Technical Reference Manual CPD-74B, Hydrologic Engineering Center, Davis, Calif, USA, 2000.
    Another reference for detailing the use of HEC-HMS might be the following:
    D. Halwatura and M. M. M. Najim, "Application of the HECHMS model for runoff simulation in a tropical catchment," Environmental Modelling and Software, vol. 46, pp. 155–162, 2013.

35. P8/L1-6: Please detail the rainfall logger units. What brand and manufacturer, is there information about accuracy? Who did collect the data or are these public domain? If so, a proper citation is needed? Who did the analysis

36. P8/L9: Please change to "coefficient of determination".

37. P8/12: Please give proper reference to the Thiessen Polygon method.

38. P8/L14: What data correction was conducted? This needs more detail or proper referencing.

39. P8/L15-20: Figure 2 in general needs better explanation. Figure 2b needs to be explained particularly, as the data plotted are maximum rainfall data (at least it says so in the figure caption). When was this rainfall event? Do maybe exist discharge measurements in some of the rivers in the catchment area? This might help detailing the hydrological situation in the catchment.

40. P8/L21: Caption – please stick to one chosen way of calling the IDF's. Earlier in the manuscript the term was hyphenated, now it is not. Also, there is no need to repeat what was introduced as an abbreviation before. Please revise.

41. P8/L23: Where are the positions of the auto-logger stations? How many?

42. P8/L25: What does that mean? Did the authors perform a peak-over-threshold method based on annual rainfall maxima? This needs more description in order to be comprehensible.

43. P9/L1: Please let the readers know more about the existing data? E.g. what time record was used to compile the existing rainfall intensity data?

44. P9/L4: What is the "Chart" station? Is this meant to be a name or something else?

45. P9/L5f: What was compared? Be more precise and specific.

46. P9/L5ff: Figure 3a shows distinct differences and the authors fail to explain those. E.g., it would be advantageous to the manuscript to detail how the differences were brought about and yet, sufficient discussion is lacking. Please revise accordingly.

47. P9/L14: What is a "usual" storm burst? Please drop mentioning or explain in detail.

48. P10/L1-3: As the manuscript claims to deal with climate change and land use changes on dam reliability, it is odd that climate change impact on the rainfall intensity is treated just by referencing to work which was done by other (external) parties (NAHRIM). The reader is presented linear factors and there is no indication where these factors came from. The authors need to detail, how the CCF were produced. Was it regional downscaling and how did this happen? Then, it need discussion how this will affect the catchment area which could be accomplished by computing the differences in discharge. The reviewer figures that this might be somewhere else in the manuscript, but here is the place to mention this and discuss it properly.

49. P10/L11: Please name the agencies and stakeholders adequately. Currently there is no way to identify who is responsible for what.
50. P10/19ff: It is inadequate to present results of imperviousness and its projected change in future in such short way. As this is presented under the methodology section, one would at least hope to learn, how the main results – an increase in imperviousness of 4.5% - had been assembled. The authors cannot assume the reader to second-guess how these values were developed.
51. P10/L20: Mismatch between Figure 3c and text. What is the year the authors try to project land use data onto? 2020 or 2030? This should be devised more carefully.
52. P11/L1-4: Reference needed.
53. P11/L8ff: Please re-arrange the manuscript structure and refrain from micro-fracturing the text with all those 3rd level captions. This reads very odd.
54. P21/L3: Insufficiently detailed reference. Please revise or dig out the proper scientific basis of the work cited.
55. P11/L10ff: Awkward grammar, need revision.
56. P11/L19: Define "storm efficiency" as well as "storm maximization".
57. P12/L4-9: The mention of the chosen reference storm event needs better citation. Also, how did the transposition of the storm event into the area of interest happened. Little is provided for the reader who might be well interested to know whether this was just by guessing or by some hidden scientific means.
58. P12/L12: Engineers don't "operate", they design, plan, optimize or manage, consider word change.
59. P12/L16-19: Grammar revision needed. Please also increase clarity of writing.
60. P12/L28-P13/L10: Although the described method might be the method of choice in Malaysia, this part of the method section needs to be tied into the international context. One would need to answer, how the conversion from PMP to PMF is conducted in other parts of the world; in particular, it needs a portion in the introduction referring to standard procedures provided in literature.
61. P13/L11f: Do the authors mean "retention potential" or why does an estimate of the retention damage is of need? My understanding is that it is necessary to know how much water is held back during a major rainfall event and with what delay does it arrive at a certain site. This to my knowledge is called retention potential.
62. P13/L24: Figure 4 is introduced but the essence of what the figure shows is not mentioned. There needs to be more and specific information and description in the manuscript text.
63. P14/L1-6: Where do the readers find all the mentioned values? There needs to be a table or explicit mentioning in the text, what parameters were chosen.
64. P14/L13-16: It would be advantageous to present input parameters used for the modelling. Please amend the manuscript and properly describe/interpret them.
65. P15/L11-24: It is paramount to discuss properly why the different methods used yield so different results. In particular, the physical method (depicted in gray) yields very large precipitation values and it is not well explained why this happens. Is there an actual physical reason why the two Hershfield methods fail to produce such large precipitation values?
66. P21/L8: Editor name needs revision.
67. P25: Left panel of Fig. 1 needs a closer view. Additionally, legend fonts are illegible, there are no city and tributary names provided. Information content is thus poor and needs improvements. The reviewer understands that some of the methods provided

in here rely on GIS, so why did the authors not use such a GIS system to provide a much clearer and precise overview figure?

68. P26: Please add inset with greater location to Fig. 2b. Plot the rain fall stations used in this study accordingly. How should the reader know where these are? This will add to the impact of the manuscript. Positioning of the green rainfall amounts in the map need adjustment as they should not sit on top of the isohyets.

69. P27: All panels of figure 3 need larger label font size and improved readability. Correct mismatch between text and figure 3c as well. It also says current (2012) whereas in the text P10/L22, the authors refer to 2015. The reviewer does not hope that this material has been published elsewhere a couple of years ago.

70. P29: Figure 5. Please use SI units to label y axis. It is assumed that this is mm precipitation, but this needs to be indicated for readability. Also, legend does mention a "hydro meteorological model (gray)" and the two Hershfield methods, but it is fairly difficult to comprehend from the manuscript what all this should be. There needs to be more explanation.

71. P30: Figure 6: Use SI units only in the manuscript, drop cumecs units and abbreviate properly. Adjust figure caption as there is no mentioning of the larger rainfall durations (5 and 7 days).

---

## Author Comment (AC1) · 1 Jun 2016

**Response to Referee's comments**

The authors thank the Referee for the comments to strengthen the presentation of our results. The manuscript was modified to respond to the Referee's issues as following descriptions:

**Reviewer #1**

Major issues:

1. Rephrasing title and omit the use of Reliability on the Manuscript and title.
   Re: The title was modified and "Reliability" was omitted at the manuscript.
2. Show the rainfall data of the last 30 years, which was used to produce figure 3.
   Re: The last 30years rainfall data in producing figure 3 was added. The IDF Curves for selected station of Malaysia were plotted that are available in MSMA 2nd Edition. 2012, DID Malaysia. The main approach is based on the Gumbel distribution.
3. Add a discussion on the required length of a data series, to capture a climate change trend. This discussion should be added to the introduction.
   RE: A discussion on required length of a data series, to capture a climate change was added to the introduction section.
4. There is a section missing about calibration and validation of the hydrological model.
   RE: The related description with the calibration process was add to the section 3.6.

Minor comments;

1. Please add a small profile of the spillway, with dimensions of the spillway crest level, embankment crest level, etc,. . .
   RE: The descriptions were added to the section 2 of the manuscript.
2. Please add a small Please add some details of the Hershfield statistical method (including envelope), and the transposition method.
   RE: The description was added to the section 3.5.2.
3. What is the spatial distance between the case study and the one where the storm occurred?
   RE: The spatial distance between the case study and the one, where the storm occurred is 450 km, where both are located at the peninsular Malaysia.
4. Introduce all acronyms in the abstract.
   RE: All acronyms were introduced in the abstract.
5. Line 4 -8,page 6:: sentence is unclear.
   RE: The sentence was re-written.
6. Line 27,page 6: introduce "RL" acronym.
   RE: "RL" was deleted form the sentence.
7. Line 15,page 7: which one was used: "by a function model of response or a catchment rainfall–runo_,"
   RE: The sentence was corrected.
8. line 22 -23, page 7, sentence is unclear.
   RE: The sentence at line 23 was deleted.
9. line 1 -2, page 8, sentence is unclear.

RE: The sentence was re-written.

10. Line 7-8, page 8,the sentence is missing a verb? maybe check?

RE: The sentence was re-written.

11. Line 6, page 8, replace "was" with "were"

RE: The sentence was corrected.

12. Line 13, page 8,the sentence is unclear.

RE: The sentence was corrected.

13. Line 10, page 10, replace "was" with "were"

RE: The sentence was corrected.

**Reviewer #2**

General comments:

1. The manuscript does not read well in technical and linguistic terms. Before publication, it is consequently necessary to have it proof-read by a native speaker. The reviewer started suggesting language and expression improvements for the abstract but stopped as it was too tedious to continue throughout the manuscript. This needs a professional service before being ready for publication.

RE: The paper was revised to improve the scientific vocabulary by the co-authors having Professorship position in the fields of Water Resources, River Engineering, Hydraulic, Geotechnical Engineering, Eco-Engineering, and Coastal Protection. Besides, the grammar of text was improved after editing by a journal expert editor.

2. The introduction only partially touches the relevant literature. I cannot see any reference to the manifold bulletins which are published by the International Commission on Large Dams (ICOLD) which is a standard resource of information in regard to dams in general. It would suit the purpose of this manuscript to show that the findings (which are much on the applied side anyway) are well in agreement with those stipulations on dam constructions.

RE: The data and bulletins content of the ICOLD are not free. Although, the authors could not reach to the contents as the library did not authorization for the ICOLD web-site, it is highly appreciated if the Journal or reviewer help us to access to the ICOLD database.

3. The introduction of the PMF is somewhat weak and deserves a more thorough description in order to allow readers understand what is spoken about later on. E.g., it is not enough to reduce the PMF to be influenced to change of land use and climate. It definitely is also a function of the drainage area, its topography and slope characteristics, hence the whole interplay of factors needs at least to be mentioned. In this context it would be interesting to also compare the author's results of conversion of PMP to PMF against empirical envelope methods such as the Creager or the Francou-Rodier equation which relate the peak flow with the drainage area. Please direct a comparison towards these empirical relations and discuss how this fits into the climate change influence discussed herein.

RE: More description was added to the manuscript about the fundamental proceeds of PMF calculating.

4. The introduction misses out on a key element in scientific papers: the level of novelty is not addressed at all. This observation aligns with the lack of clarity in terms of objectives and goals which are not found to be mentioned in the introduction. It is required to more clearly state what objectives the authors pursued and where those objectives are tied into the lack of knowledge which ideally was found from a proper literature review/discussion. All those elements I cannot find easily in the introduction and thus it needs significant improvements.

RE: More descriptions were added to the introduction of the manuscript.

5. Besides all the issues regarding the methodology section provided below, there is one main issue the reviewer holds against the authors. It relates to the modelling of the flow in the vicinity of the dam site which usually is composed of overflow conveyed over the spillway and the flow released through the sluice gate at the bottom of the dam. It does not become clear why there was no more modelling involved other than the parametric model HEC-HMS aiming at simulating the system on the catchment scale. In particular the flow through the spillway requires more sophisticated means of simulation, either 1D with e.g. HEC-RAS, or given the steep slopes of many spillways worldwide, better 2D or 3D models linked together in a model cascade.

RE: More description about the spillways and the discharge analyses and graph were added to the manuscript. The overtopping model is in line with referenced literature and the manual of the HEC-HMS.

6. It is also important to look into backwater effects which might occur downstream of the dam site effectively reducing the flow capacity of the spillway and river crosssection involved. The current work does not convince that there has been enough focus on those local effects which however become crucial when looking into dam reliability. And again, there is no information regarding the actual dam structure which

makes it extremely difficult to judge what is going on there. The authors need to provide a fair bit of information to let the audience value the work by themselves.

RE: The Red Hill dam, spillways and reservoir features were indicated in tables1, 2 and 3 in the manuscript. The dam Spillways are ogee type which the discharge characteristics spillway can be derived from the characteristics of the spillway including the coefficient of discharge, the head of water on the spillway crest and the width of the spillway.

The values of the coefficient of discharge is influenced by a number of factors such as (1) the relation of the actual crest shape to the ideal nappe shape, (2) the depth of approach, (3) the inclination of the upstream face, (4) the contraction caused by the crest piers and abutment, (5) the interference due to downstream apron, and (6) *the submergence of the crest due to downstream water level*, which have been studied and well documented by USBR.

The effect due to interference caused by the downstream apron, in terms of variation of (Ca/C), the ratio of modified coefficient to free discharge coefficient,

with respect to apron effect, has been studied in terms of variation of the parameter $(h_d+d)/H_e$. where These parameters are shown in the following figure.

[Figure]

It was found that when this parameter exceeds about 1.7, the downstream floor has practically no effect on the discharge coefficient (Khatsuria, 2015)[1].

The effect of submergence of crest by Tail Water (TW) on discharge coefficient is studied in terms of parameter $(h_d/H_e)$, also called degree of submergence. It was reported that the coefficient of discharge begins to be influenced by the submergence when the degree of submergence (expressed as the depth of TW above the crest/head on crest) exceeds about 40% (Khatsuria, 2015).

According to the design and operation data, no record was found to show that TW above the crest/head on crest exceeds about 40%. Therefore, for the Red Hill dam, there is no downstream effect on spillways discharge capacity. A description was added to the section 2 of the manuscript, briefly.

7. The result and discussion section is particularly disappointing to read. From what is promised in the manuscript title one would expect to learn how the climate change and land use variation over the years would contribute to increase PMF used for the design of a dam as an application. But there is no more distinguishing between the two factors the authors set out to focus on. From the title, readers would expect to know how a single change, say of climate only, would affect the design for a dam. What is more, the manuscript suffers greatly from the unclear language in which it is written. There is a great need to be more specific and it is advised to use as many references as possible to underpin the author's case.

8. 8. Section "Conclusion" should either be renamed to "Summary" or there needs to be actual meaningful conclusions to be drawn which will eventually help others to learn how to address climate and land use change with respect to dam reliability.

   RE: based on the first major comment for the first reviewer the title of the manuscript was modified. Consequently, the result and discussion section was improved in line with the title.

9. Moreover, it does not seem convincing that the maximum water level inside the impoundment just marginally increases and that the dam is still safe given the drastic
* * *
[1] Khatsuria, R. M. (2004). *Hydraulics of spillways and energy dissipators*. CRC Press.

increase in rainfall detailed in the methodology section. It might be more important to vary the climate change factor quite a bit more in order to see how these variations will affect the water levels inside the impoundment. Also, there is a need to look into the processes where the spillway is involved.

RE: Please refer to the response to the first minor comment of the first reviewer and response to comment no. 6 of the second reviewer.

More detailed comments (keyed to page (P) and line (L) numbers):

1. P3/L9: Wrong/imprecise wording "Flood rises". Consider words like flow depth, surface elevation, flood level.
   RE: The sentence was revised.
2. P3/L9: Please write "Considering the climate change factor …"
   RE: The sentence was revised.
3. P3/L11: Please revise to "…, the year of 2020…"
   RE: The sentence was revised.
4. P3/L14: Remove "of" before 2030
   RE: The sentence was revised.
5. P3/L16: Define PMP, PMF, HEC-HMS also in Abstract.
   RE: PMP, PMF, and HEC-HMS was described in Abstract.
6. P3/L16: Please modify to: "The software HEC-RMS…"
   RE: Although the applied software was HEC-HMS, the sentence was revised by adding "software". The HEC-HMS file of the project is available up on a request.
7. P3/L17: "flood rises", please see above comment.
   RE: The sentence was revised.
8. P3/L18: Mention what "modified technique was used" or remove from abstract.
   RE: The sentence was revised.
9. P3/L20: Use "investigated" instead of performed.
   RE: The sentence was revised.
10. P3/L23: What is "marginally adequate", reads awkward.
    RE: The sentence was revised.
11. P3/L25: Please use: "…, the dam safety in terms of hydrology was assured."
    RE: The sentence was revised.
12. P4/L8: "As the importance of safety increases…" Please focus this sentence more clearly, it reads very unspecific. Without changes in external forcing such as climate change, there would definitely be no reason to re-assess dam safety as dams are usually build according to well-drafted design codes which include measures for safe constructions even under extraordinary loads.
    RE: The sentence was revised.
13. P4/L19: Please add the exact reason why a re-assessment is necessary. This does not become clear without consulting the reference.
    RE: More description with a reference was added to the manuscript
14. P4/L22: "Spillways are a common way to…"

RE: The sentence was revised.

15. P4/L23: Consider reformulating: "If a spillway is not designed properly…"

RE: The sentence was revised.

16. P4/L26: Please enclose abbreviations in brackets.

RE: The sentence was revised.

17. P4/L 27: "The body of a dam…" The actual design criteria needs to be explained in more detail. In particular, it is not about the body of the dam, but about the amount of water that can be stored in the dam impoundment and conveyed through the overflow spillway such that no crucial part of the dam construction is affected. This needs to be detailed in a more technically correct way.

RE: The sentence was revised.

18. P5/L1: What is normal in the context of historical climate data? This statement needs revision. Also, the sentence requires more explanation on what the "targeted performance" is. It appears as if this is a portion of some other work the first author led, but it is difficult to understand without further explanation. E.g., it would be advantageous to learn, for which types of infrastructure these statements were made and whether the methodology is applicable to dams in general and earth filled dams in particular.

RE: The sentence was amended as adding more details was lengthening the introduction section.

19. P5/L17: Please detail the term "mismanagement" in this context. The term includes a pre-judgement and needs additional justification.

RE: The sentence was revised.

20. P5/L20: Refrain from stating the obvious yet highly contested without properly referencing.

RE: The sentence was revised.

21. P5/L20f: Sentence difficult to understand, please revise.

RE: The sentence was revised.

22. P5, L21: Incorrect grammar (tenses mixed).

RE: The sentence was revised.

23. P6/L4: Grammar issue "It …"

RE: The sentence was revised.

24. P6/L25: What height is referred to? Crest height? Please be specific. Is there a local datum to which this is referred to? Please state the important facts.

RE: The sentence was revised. More details of the structures were added to the manuscript.

25. P6-Fig1: Figure is neither mentioned nor explained. Please tie in the Figure and introduce the study area to the readers.

RE: Description of figure 1 was added to the manuscript.

26. P7/L1: What is "RL", please always write in full the abbreviation used.

RE: The sentence was revised.

27. P7/L1: What is "Integrated Agricultural"? Unclear.

RE: The sentence was revised.

28. P7/L3: "… double cropping planting …" Unclear what this sentence should say? Needs revisions.

RE: The sentence was revised.

29. P7/L8: Please always use SI units and its derivatives. I assume this is m^3/s?

RE: The unit was revised.

30. P7/L9: "the GIS tool …" What GIS tools were used and what methods were applied to

yield the information stated? Please be precise in your description of the work.

RE: The sentence was revised.

31. P6/L7: Section 2 requires more visual details, e.g. provide a plan view of the dam and impoundment. Detailed overview over the spillway and related measures taken at the dam site would also greatly help the reader understand how the spillway is constructed as its cross-sectional area, slope, energy dissipation mechanisms will most certainly affect how well storm water discharge is conveyed downstream. This is essential to this study which in my understanding exactly addresses this questions under climate change aspects.

RE: More details of dam and spillways were added to the manuscript.

32. P7/L19: "fundamental PMP convention". Please detail, unclear. Needs citation/reference.

RE: As the process has been describe in detail in sub-sections, the sentence was amended

33. P7/L22: Please properly cite the software the authors used. USACE, "Hydrologic modeling system," HEC-HMS Technical Reference Manual CPD-74B, Hydrologic Engineering Center, Davis, Calif, USA, 2000. Another reference for detailing the use of HEC-HMS might be the following: D. Halwatura and M. M. M. Najim, "Application of the HECHMS model for runoff simulation in a tropical catchment," Environmental Modelling and Software, vol. 46, pp. 155–162, 2013.

RE: The reference was cited.

34. P8/L1 -6: Please detail the rainfall logger units. What brand and manufacturer, is there information about accuracy? Who did collect the data or are these public domain? If so, a proper citation is needed? Who did the analysis?

RE: More description was added to the manuscript.

35. P8/L9: Please change to "coefficient of determination".

RE: The sentence was revised.

36. P8/12: Please give proper reference to the Thiessen Polygon method.

RE: The reference was cited.

37. P8/L14: What data correction was conducted? This needs more detail or proper referencing.

RE: The reference was cited.

38. P8/L15-20: Figure 2 in general needs better explanation. Figure 2b needs to be explained particularly, as the data plotted are maximum rainfall data (at least it says so in the figure caption). When was this rainfall event? Do maybe exist discharge

measurements in some of the rivers in the catchment area? This might help detailing the hydrological situation in the catchment.

RE: More description was added to the manuscript.

39. P8/L21: Caption – please stick to one chosen way of calling the IDF's. Earlier in the manuscript the term was hyphenated, now it is not. Also, there is no need to repeat what was introduced as an abbreviation before. Please revise.

RE: The title was revised.

40. P8/L23: Where are the positions of the auto-logger stations? How many?

RE: Please refer to the response of the comment no. 34.

41. P8/L25: What does that mean? Did the authors perform a peak-over-threshold method based on annual rainfall maxima? This needs more description in order to be comprehensible.

RE: Due to limitation of word number the description of the Gumble Distribution Method was referred to its citation in the manuscript. The description are available up on a request.

42. P9/L1: Please let the readers know more about the existing data? E.g. what time record was used to compile the existing rainfall intensity data?

RE: the comment was not clear and the related words were not found at P9/L1. More clarification is appreciated.

43. P9/L4: What is the "Chart" station? Is this meant to be a name or something else?

RE: That is a name of the station.

44. P9/L5: What was compared? Be more precise and specific.

RE: The sentence was revised.

45. P9/L5: Figure 3a shows distinct differences and the authors fail to explain those. E.g., it would be advantageous to the manuscript to detail how the differences were brought about and yet, sufficient discussion is lacking. Please revise accordingly.

RE: More description was added to the manuscript.

46. P9/L14: What is a "usual" storm burst? Please drop mentioning or explain in detail.

RE: The sentence was revised.

47. P10/L1 -3: As the manuscript claims to deal with climate change and land use changes on dam reliability, it is odd that climate change impact on the rainfall intensity is treated just by referencing to work which was done by other (external) parties (NAHRIM). The reader is presented linear factors and there is no indication where these factors came from. The authors need to detail, how the CCF were produced. Was it regional downscaling and how did this happen? Then, it need discussion how this will affect the catchment area which could be accomplished by computing the differences in discharge. The reviewer figures that this might be somewhere else in the manuscript, but here is the place to mention this and discuss it properly.

RE: More description was added to the manuscript.

48. P10/L11: Please name the agencies and stakeholders adequately. Currently there is no way to identify who is responsible for what.

RE: The agency was corrected.

49. P10/19: It is inadequate to present results of imperviousness and its projected change in future in such short way. As this is presented under the methodology section, one would at least hope to learn, how the main results – an increase in imperviousness of 4.5% - had been assembled. The authors cannot assume the reader to second-guess how these values were developed.

RE: More description was added to the manuscript.

50. P10/L20: Mismatch between Figure 3c and text. What is the year the authors try to project land use data onto? 2020 or 2030? This should be devised more carefully.

RE: The year was corrected.

51. P11/L1 -4: Reference needed.

RE: The reference was cited.

52. P11/L8: Please re-arrange the manuscript structure and refrain from micro-fracturing the text with all those 3rd level captions. This reads very odd.

RE: The manuscript captions was re-arranged in 2 level numbering.

53. P21/L3: Insufficiently detailed reference. Please revise or dig out the proper scientific basis of the work cited.

RE: The reference was corrected.

54. P11/L10: Awkward grammar, need revision.

RE: The sentence was revised.

55. P11/L19: Define "storm efficiency" as well as "storm maximization".

RE: Due to limitation of word number the description of the storm maximization is referred to Casas et al. (2008). The description are available up on a request.

56. P12/L4-9: The mention of the chosen reference storm event needs better citation. Also, how did the transposition of the storm event into the area of interest happened. Little is provided for the reader who might be well interested to know whether this was
just by guessing or by some hidden scientific means.

RE: As the transposition is step by step procedure the reference was cite if reader might be interested to find more details.

57. P12/L12: Engineers don't "operate", they design, plan, optimize or manage, consider word change.

RE: The sentence was revised.

58. P12/L16-19: Grammar revision needed. Please also increase clarity of writing.

RE: The sentence was revised.

59. P12/L28-P13/L10: Although the described method might be the method of choice in Malaysia, this part of the method section needs to be tied into the international context. One would need to answer, how the conversion from PMP to PMF is conducted in other parts of the world; in particular, it needs a portion in the introduction referring to standard procedures provided in literature.

RE: More details were referred to the cited references in the manuscript.

60. P13/L11: Do the authors mean "retention potential" or why does an estimate of the retention damage is of need? My understanding is that it is necessary to know how much water is held back during a major rainfall event and with what delay does it arrive at a certain site. This to my knowledge is called retention potential.

RE: The sentence was revised.

61. P13/L24: Figure 4 is introduced but the essence of what the figure shows is not mentioned. There needs to be more and specific information and description in the manuscript text.
    RE: Table 2 was added to show the calculated physical parameters of the study area.
62. P14/L1-6: Where do the readers find all the mentioned values? There needs to be a table or explicit mentioning in the text, what parameters were chosen.
    RE: Table 2 was added to show the calculated physical parameters of the study area.
63. P14/L13-16: It would be advantageous to present input parameters used for the modelling. Please amend the manuscript and properly describe/interpret them.
    RE: The related description with the calibration process was add to the section 3.6.
64. P15/L11 -24: It is paramount to discuss properly why the different methods used yield so different results. In particular, the physical method (depicted in gray) yields very large precipitation values and it is not well explained why this happens. Is there an actual physical reason why the two Hershfield methods fail to produce such large precipitation values?
    RE: Due to limitation of word number more details can be found at the references cited at the manuscript. The description are available up on a request.
65. P21/L8: Editor name needs revision.
    RE: Editor name was revised.
66. P25: Left panel of Fig. 1 needs a closer view. Additionally, legend fonts are illegible, there are no city and tributary names provided. Information content is thus poor and needs improvements. The reviewer understands that some of the methods provided in here rely on GIS, so why did the authors not use such a GIS system to provide a much clearer and precise overview figure?
    RE: The figure was improved.
67. P26: Please add inset with greater location to Fig. 2b. Plot the rain fall stations used in this study accordingly. How should the reader know where these are? This will add to the impact of the manuscript. Positioning of the green rainfall amounts in the map need adjustment as they should not sit on top of the isohyets.
    RE: The figure was improved.
68. P27: All panels of figure 3 need larger label font size and improved readability. Correct mismatch between text and figure 3c as well. It also says current (2012) whereas in the text P10/L22, the authors refer to 2015. The reviewer does not hope that this material has been published elsewhere a couple of years ago.
    RE: The figure was improved. The year was corrected
69. P29: Figure 5. Please use SI units to label y axis. It is assumed that this is mm precipitation, but this needs to be indicated for readability. Also, legend does mention a "hydro meteorological model (gray)" and the two Hershfield methods, but it is fairly difficult to comprehend from the manuscript what all this should be. There needs to be more explanation.
    RE: The figure was improved.

70. P30: Figure 6: Use SI units only in the manuscript, drop cumecs units and abbreviate properly. Adjust figure caption as there is no mentioning of the larger rainfall durations
(5 and 7 days).
RE: The figure was improved.

Regarding the mentioned modifications, revised paper including improved sections and descriptions will be forwarded to the editorial panel of the HESS journal to be considered before final publishing.